# Alley Cropping Mitigates the Impacts of Climate Change on a Wheat Crop in a Mediterranean Environment: A Biophysical Model-Based Assessment

Francesco Reyes [1,2,*], Marie Gosme [1], Kevin J. Wolz [1,3], Isabelle Lecomte [1] and Christian Dupraz [1,*]

1 National Research Institute for Agriculture, Food and the Environment (INRAE), University of Montpellier, 34060 Montpellier, France; marie.gosme@inrae.fr (M.G.); isabelle.lecomte@inrae.fr (I.L.)
2 Dipartimento delle Culture Europee e del Mediterraneo: Architettura, Ambiente, Università degli Studi della Basilicata, 75100 Matera, Italy
3 Savanna Institute, Madison, WI 53715, USA; kevin@savannainstitute.org
* Correspondence: francesco.reyes@unibas.it (F.R.); Christian.dupraz@inrae.fr (C.D.)

**Abstract:** Introduction: Climate change (CC) and the increased occurrence of extreme climatic events pose a serious threat to crop yields and their stability worldwide. This study analyzed the CC mitigation potential of an alley cropping system on crop physiological stresses and growth as compared to a monoculture system. Materials and Methods: Growth of winter durum wheat, cultivated alone (agriculture) and in combination with hybrid walnut (agroforestry), was simulated with the Hi-sAFe agroforestry model, as driven by business-as-usual Intergovernmental Panel on Climate Change (IPCC) projections, split into three scenarios, representing *Past* (1951–1990), *Present* (1991–2030), and *Future* (2031–2070) climatic conditions. Crop growth and the occurrence of thermal, nitrogen, and water stresses were analyzed. Results: Cold-related stresses were modest in *Past* and almost disappeared over time. Heat, drought, and nitrogen stresses increased about twofold from *Past* to *Future*, but were reduced by 20–35% in agroforestry, already with medium-sized trees (diameter at breast height (DBH) of about 10–15 cm). Crop yields in agriculture increased from *Past* to the end of *Present* and then remained stable. This moderately decreased with tree age in agroforestry (especially in *Future*). Discussion: The impact of CC on the crop was buffered in agroforestry, especially for the most extreme climatic events. The mitigation of crop microclimate and the increased stability of crop yields highlight the potential of agroforestry as a CC adaptation strategy.

**Keywords:** agroforestry; alley cropping; silvoarable; crop; tree; wheat; walnut; water stress; thermal stress; nitrogen stress



## 1. Introduction

Plant growth and thus the agricultural sectors are sensitive to both causes (variation in greenhouse gases concentrations) and consequences (increases in air and soil temperatures, changes of quantity and quality of the incoming direct solar radiation, of total rainfall, and their temporal distributions) of climate change (CC) worldwide [1]. One primary factor affecting crop yield is air temperature. Worldwide minimum and maximum daily temperatures during the growing seasons increased on average by 0.3 and 0.2 °C per decade, respectively, from 1980 to 2011 [2]. Global warming is expected to increase at high rates, especially in the Mediterranean area, with temperatures higher than the global average (+20%), especially in summer months (+50%) [3]. Whether a crop may benefit or not of such a change depends on whether the site-specific microclimate will increasingly meet optimal crop growth temperatures at each development stage (e.g., by reducing frost risk in the cold season, while increasing respiration costs and thus carbon losses or the occurrence of drought stress in the hot season) [2]. Temperature increases have been related to crop yield losses in barley, maize, wheat, sorghum, soybean, and rice, with projections

suggesting yield reductions of up to 70% by the end of the century [2,4,5]. Temperature increase has also been suggested to be the cause of long-term yield stagnation in Europe despite continuing efforts to improve cultivars [6].

Precipitation patterns [7] have major impacts on soil available water, and thus on drought, drought induced crop stresses, and photosynthesis. Increases in precipitation have been projected at both equatorial and high latitudes, but along with much higher evapotranspiration rates. Precipitations are projected to decrease at tropical and temperate latitudes [8]. Summer and spring rainfalls are expected to be reduced by up to 30% by the late twenty-first century [9,10] in Southern European countries and even further in North African ones, with a concurrent significant increase in rainfall intensity in autumn and winter months [11]. Prolonged droughts and extreme precipitations are expected as a consequence of more stable atmospheric pressure, induced by changes in patterns of high-altitude streaming. The increased intensity of rain events may reduce soil moisture content because of increased runoff, changes in timing of soil water recharge, and, indirectly, soil erosion. The combination of temperature and rainfall changes has significantly increased drought extent and severity, especially in the Mediterranean area, Africa, East Australia, and South and East Asia since 1970 [12–14].

Rising atmospheric carbon dioxide concentration is generally expected to positively impact plant water use efficiency and productivity [15]. However, this may not be the case in situations where other plant growth factors, such as nitrogen and phosphorus, limit plant growth [16].

The effects of environmental variables on microclimate and resource availability, and thus on plant growth and agricultural system performances, are mediated by characteristics of the system design, such as the planted species, their distribution in the agricultural plot, and their management. For example, crop planting dates can be modified to adapt to CC [2,17]. In agroforestry systems (AF), woody vegetation (trees or shrubs) is deliberately integrated with crop and/or animal production, with the aim of providing ecological and economic benefits [18]. Among AF practices, alley cropping refers to the growing of arable or herbaceous crops in wide alleys defined by rows of trees and possibly shrubs [19–23]. Alley cropping has been shown to increase the overall productivity and resilience of agricultural systems [24] and is a promising new land-use type in Europe for CC mitigation (e.g., carbon storage [25–27]).

The key drivers of CC adaptation in alley cropping include:

- Improved organic matter content, soil structure and stability, water infiltration and field capacity, reduced erosion [19,23];
- Increased available nitrogen and improvement of nitrogen cycling, and thus reduced risk of nitrogen deficiency and leaching [19,22,23];
- Increases in the overall light use efficiency [22];
- Alleviation of temperature and water stresses on crops due to mitigation of the crop microclimate (reduced radiation, reduced air and soil temperature, increased air humidity, reduced wind velocity) [28,29];
- Slowdown in the crop phenology that may allow to capture more light and nitrogen by lengthening the growing season [30,31].

However, the integration of trees with crops may also create less favorable growth conditions. Reduced light availability may reduce photosynthesis and lead to significantly lower yields, especially in C4 plants [19]. The choice of species with overlapping root systems, throughout development or at some specific stages, and similar temporal patterns of water demand, may lead to enhanced competition for water and nutrients. In addition, AF-related management may be more complex than in monocrop systems, as a larger variety of field operations may be needed.

In the Mediterranean region, climate change is expected to result in increased risk of heat stress and drought [32]. The mitigation effect that trees might have on these risks is uncertain: although trees can reduce air and crop temperature, the effect on crop phenology might lead to slower development and so a delay in the most sensitive stage, i.e., anthesis

for cereals [33], which could happen when temperatures are even higher. For the mitigation of drought risk, although trees can reduce atmospheric evaporative demand [34], they also compete for water with crops, despite the partial vertical segregation of crop and tree roots [35]. Therefore, whether the use of alley cropping systems may be an appropriate CC adaptation strategy in Mediterranean environments is an open question [30,31], although some Mediterranean alley cropping systems have been successful for thousands of years, e.g., olive trees intercropped with cereals [28]. The success of AF systems in mitigating detrimental effects of CC on plant growth is related to the suitability of specific assemblies of species to the given environments. However, long-term field experiments studying the effect of climate change on agroforestry systems are very difficult to carry out due to the size of the system and the duration of tree growth: climate manipulation over mature trees for decades is costly and time-consuming [20,36–38]. Furthermore, to disentangle the effect of different processes explaining the effect of agroforestry on climate change adaptation, it is necessary to access intermediary variables whose field assessment is considered unaffordable (e.g., several physiological stresses at daily time step).

In this context, process-based numerical models become essential tools to perform in-silico experiments, possibly including some randomization in the inputs to obtain replicated simulation outputs, and analyze the possible effects of CC on the systems. Process-based models may also allow evaluation of facilitation and complementarity relationships among plants by integrating plant interactions with microclimatic conditions and resource availability. Several AF models have been described in the literature, differing in scope and complexity [39]. However, trees and tree–crop interactions are rarely included in modeling efforts assessing the impact of CC [40]. The Hi-sAFe model has been specifically designed to represent AF plots, simultaneously taking into account resource (light, water, and nitrogen) use by trees and crops and the heterogeneous microclimatic effects of trees on the crops [41,42].

The aim of this study is to assess the influence of an alley cropping system in terms of crop growth conditions, with respect to CC in a Mediterranean environment. We chose an AF system composed of a commercial timber tree species with delayed phenology—hybrid walnut trees (*Juglans × intermedia* (Carr.))—in combination with a winter cash crop—durum wheat (*Triticum durum*). This mixture is potentially suitable, given the complementarity in phenology and use of light of the two species [43]. Crop stresses and productivity of the alley cropping system and of the wheat monocropping system (A for agriculture) were simulated over the long-term (40-year cycles) by means of the Hi-sAFe model, and as driven by Intergovernmental Panel on Climate Change (IPCC) climate projections.

## 2. Materials and Methods

### 2.1. Introduction to the Case Study

The Hi-sAFe model [41,42] was used to simulate the effect of theoretical climate scenarios using, as a case study, a real plot over three climatic scenarios representing *Past*, *Present*, and *Future* climatic conditions. The plot is located in Southern France (Restinclières, 43.70412° N, 3.86152° E, 62 m ASL, parcel A2), where agroforestry, forestry, and agricultural parcels were established in 1995 and monitored since then by INRAE (INRAE UMR SYSTEM 2015). This particular plot was chosen because it was used to calibrate the model [42], and because the soil has been well characterized, down to a depth of 9.4 m, i.e., down to the bedrock [44]. Tree line orientation was east-west in the AF system. Tree spacing was 13 m between rows and 9 m within the row (85 trees/ha) in AF, as representing the mean tree density obtained in the field after selective tree thinning, applied nine years after tree planting. A 1-m-wide strip along the tree lines in AF was covered with unmanaged Gramineae grasses, dying off about 1 month after wheat harvest. This land use cover resulted in 0.92 ha of crop per hectare of the full system.

### 2.2. The Hi-sAFe Model

Hi-sAFe (version 3.5) [41] is a biophysical, mechanistic model initially developed by the European Silvoarable Agroforestry for Europe (SAFE) project. This is an exploration tool meant to synthesize our understanding of tree–crop interactions and to predict the biophysical properties of agroforestry systems in temperate zones. The model allowed us to simulate tree–crop competition for three key resources: light, water, and nitrogen. It includes plant plasticity as a consequence of competition with 3D canopy and roots. The impacts of the modification of the crop microclimate by the trees are also modeled. The HisAFe model allows to explore the behavior of AF systems in various soil, climate, and latitude conditions, and to explore the impact of different system designs (tree and crop species, tree density, cardinal alignment of tree rows, tree spacing, etc.) and management options (e.g., tree pruning, tree root pruning, crop management practices) [41,42]. The dynamics of tree–crop interactions are calculated at a daily time step, and possibly for long durations. The model has been used to test the effect of latitude on the best tree row orientation to maximize light transmission to the crop [45].

### 2.3. Modeled System Description

Two systems were modeled with Hi-sAFe:

- Monoculture winter durum wheat (Claudio variety, agriculture: "A");
- Alley cropping of wheat with walnut trees and grass along the tree lines (agroforestry: "AF").

The AF-modeled scene in Hi-sAFe was virtually replicated in every direction using toric symmetry, approximating an infinite stand without edge effects.

The wheat crop was sown every year on the same day of the year (DOY) 210, at the same density (250 seeds m$^{-2}$) in A and AF. Soil was tilled at thirty-centimeters depth (DOY 299) and received three nitrogen applications of 30, 50, and 80 kg ha$^{-1}$ between early February and April. The days of fertilization (occurring on DOYs 34, 54, and 104 in year 1951) were linearly adjusted over time, taking into account the effect of increasing winter temperatures on earlier plant emergence across years. Accordingly, fertilizations were postponed by about one day every three years. Low branches of the trees were pruned every two years up to four meters height.

### 2.4. Climatic Data Series

Climatic projections for our experimental site were obtained from the Clipick website [46]. These contained the daily variables needed to run simulations with the Hi-sAFe model: Maximum and minimum temperature, maximum and minimum relative humidity, sum of global radiation, sum of precipitation, and mean wind speed at the daily time step. Clipick is an online tool providing climatic projections produced by global climate models proposed by the Intergovernmental Panel on Climate Change (IPCC), and down-scaled by a regional climate model [46]. In this, climatic data resulting from the Representative Concentration Pathways 8.5 scenario (*RCP8.5*, Assessment Report 5) are available, of which the historical data (*Hist*, years 1951–2005) had been evaluated against observed data, before being released via the Clipick web service [47]. *RCP8.5* is a business-as-usual scenario (with GreenHouse Gas—GHG-emission resulting from continuously growing population, little convergence between high and low income countries, modest improvements in energy intensity, and low-carbon technologies) leading to a radiative forcing of 8.5 W/m$^2$ by the end of the century [48] and likely occurring in case no major changes would happen in the drivers of CC [49]. In order to represent CC over time, we made use of the *Hist* and the RPC8.5 scenarios (years 2006–2070). Although *Hist* data (ending in 2005) could have been merged with empirical data to cover the time until present, this choice was avoided in order to keep the consistency among the data used in the scenarios, and thus of their impacts on the crops, rather than providing higher coherence with the study site for a particular scenario. The two data series obtained were thus merged and then divided into *Past* (1951–1990), *Present* (1991–2030) and *Future* (2031–2070) climatic scenarios.

Climatic series represent squared areas of 11 km side and therefore do not perfectly fit any specific location [47]. In our case, some adjustments were required to adapt the climatic series to the study site (Supplementary Materials Figure S1). In particular, we adjusted discrepancies in temperature between the years of the historical dataset overlapping with field measurements (from year 1995 to year 2014). The same correction was then applied to adjust temperatures for the *Past*, *Present*, and *Future* scenarios. We also added the water table depth using a simple empirical model predicting it as a function of rainfall of the previous days, which was calibrated based on piezometer measurements, and air carbon dioxide concentration (considering a constant concentration, to focus the study on microclimate rather than carbon dioxide effect).

### 2.5. Replication in the Virtual Experiment

Clipick does not contain any stochastic seed, and only one time series is available for the *Hist* and *RCP* climate scenarios. However, the use of a single data series in simulations of perennial plant development might overemphasize the impact of unusual successions of meteorological events. Indeed, the plant response to the environment is sensitive to the detailed sequence of climatic events. To mitigate the potential impact of unusual meteorological successions on simulations, we pseudo-replicated the data series by randomly shuffling the years of the downloaded (reference) meteorological data series. To keep climatic trends intact, shuffling was performed within 10-year windows.

Eleven reshuffled time series were generated for each climatic scenario (*Past, Present, Future*). This resulted in 66 simulations of 40 years each (two systems * three climatic scenarios * 11 replicates).

### 2.6. The Simulated Climate Change (IPCC Scenario RCP8.5)

The daily meteorological variables driving the model are synthetized in Figure 1. The climate in Restinclières is predicted to become warmer, slightly dryer, and with relatively higher incident radiation, as related to a decrease in cloudiness (Figures 1 and S2). Tmin and Tmax are predicted to increase by about 3 °C between 1951 and 2070 (respectively from 5.4 to 8.8 °C and from 19.3 to 22.5 °C), and G.Radiation by about 5% (from 14.8 to 15.6 MJ/m$^{-2}$). RHmin and RHmax would decrease by only 2% across the whole period. The total annual rainfall is expected to increase by a modest 4% (from 1017 to 1059 mm) between 1951 and 2070. However, the intensity of extreme rainfall events is expected to increase, with the largest daily rainfall rising up by 27% (from 72 to 91 mm) (Figure 1). Finally, the mean daily wind speed is expected to remain stable, but with higher interannual variability in *Future* than in previous scenarios.

Variations in Tmin, Tmax, and in most of G.Radiation and RHmin were steady across scenarios, while high fluctuations across decades were present in RainfallTot, RainfallMax, Wind, and RHmin (Figure 1). The fluctuations of the different variables were associated with each other. High values in RainfallTot were associated with high values in RHmin and Wind, and with low values in G.Radiation (with the exception of a period, centered around year 25 of *Past*, characterized by a limited amount of rainfall concentrated in few intense rain events) (Figrues 1 and S3).

The occurrence of spring and summer days with extremely high temperatures has increased over time (Figure S4—above). This includes spring days with temperatures higher than 25 °C, potentially dangerous for grain filling in wheat [6]. Temperatures below zero and frost risk (below −5 and −10 °C) in March and April, on the contrary, decreased by more than 50% from *Past* to *Present*, and almost disappeared in *Future*.

Regarding changes in the distribution and frequency of rainfall (Figure S4—below), the least affected season seemed to be autumn, with a moderate increase in heavy rainfall (>16 mm). In winter, light rainy events (<4 mm) were predicted to occur less frequently over time, while moderate (>4 and <16 mm) and extreme rains would become more frequent. Conversely, in both spring and summer, light and moderate rains were expected

to decrease, while heavy rains were expected to increase in *Present* and *Future* in respect to *Past*.

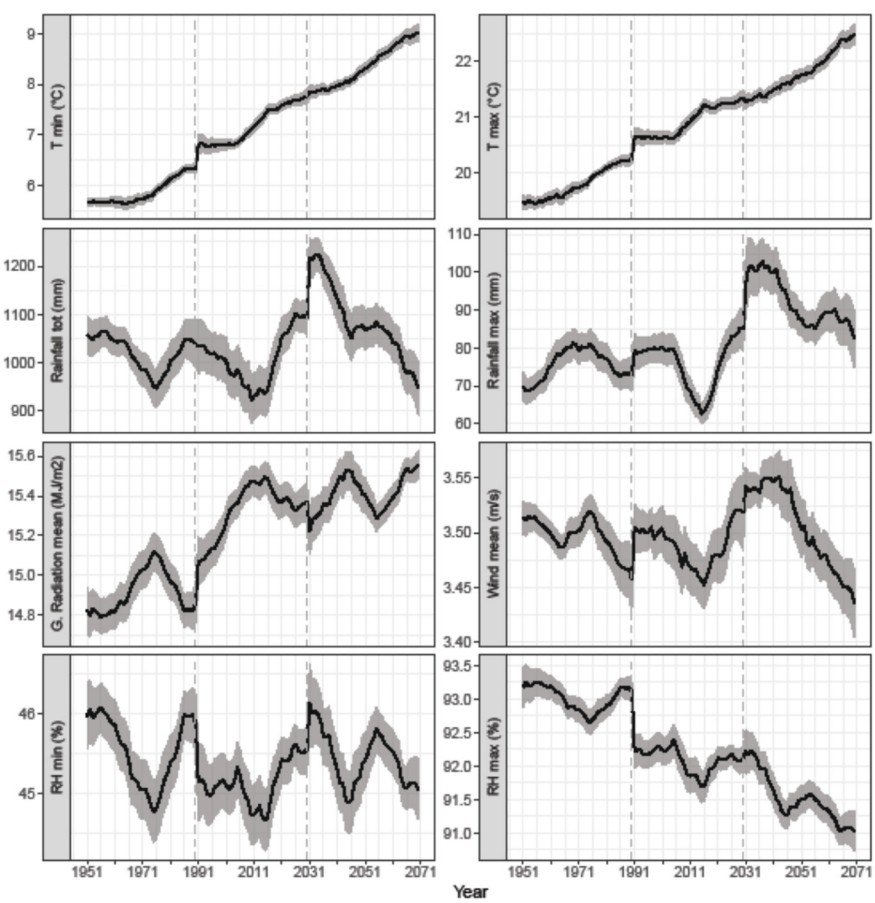

**Figure 1.** Predicted trends in annual temperature, relative humidity, rainfall, wind speed, and radiation over the simulated period. Mean of the minimum and maximum daily relative humidity and temperature; total annual and largest annual daily rainfall, mean daily mean wind speed, and sum of global radiation. Vertical dashed lines separate time in periods corresponding to the simulated scenarios. Discontinuities between periods are due to the fact that moving averages are applied to individual series of forty years. Vertical lines represent the 95% confidence interval.

### 2.7. Computation of Indices to Characterize the Occurrence and Impact of Crop Stresses

We analyzed the suitability of microclimatic conditions for crop growth across agricultural systems and scenarios in terms of available photosynthetically active radiation (PAR) and 11 crop stress variables computed by the model (Tables 1 and 2), indicating possible limitations in plant development or anticipated senescence due to non-optimal temperature, water, and nitrogen availability for the crop.

The impact of excessively cold or hot temperatures was considered by five variables. Regarding freezing temperatures, we considered the damage these could cause during plants' early growth stages in terms of loss of whole plants, and thus reduction of plant density, and to loss of leaf area (respectively the FrostStressPlantDensity and FrostStressFoliage variables), and in terms of reduction in the viability of plant flowers and fruits (FrostStressReprod). This last variable indicated no impact at any time and is not presented. Regarding hot temperatures, their impact in terms of inhibition of grain filling was considered (TempStressGrainFilling). In addition, the reductions in crop photosynthetic efficiency, due to excessively cold or hot temperatures, were considered separately by splitting the unique TempStressLue variable in two (LowTempLue, HighTempLue), based on whether the mean crop daily temperature was below or above the optimal temperature range for photosynthesis.

**Table 1.** Thermal, water, and nitrogen stresses active on the crop in the Hi-sAFe model.

| Stress Category | Stress Name in STICS | Stress Name in Hi-sAFe | Description | Further Analyzed (Yes/No) |
|---|---|---|---|---|
| Thermal | FTEMP | TempStressLUE | reduction in radiation use efficiency | Y |
| | FTEMPREMP | TempStressGrainFilling | temporary inhibition of grain filling | Y |
| | FGELLEV | FrostStressPlantDensity | reducing plant density | Y |
| | FSTRESSGEL | FrostStressFoliage | faster leaf senescence | Y |
| | FGELFLO | FrostStressReprod | reducing plant flowers and fruits | N |
| Water | SWFAC | StomatalWaterStress | reduced radiation use efficiency and transpiration | Y |
| | SENFAC | SenescenceWaterStress | faster leaf senescence | Y |
| Nitrogen | INNLAI | NitrogenLAIStress | delayed plant development and slower leaf growth | Y |
| | INNS | NitrogenBiomassStress | crop light use efficiency | N |
| | INNSENES | NitrogenSenescenceStress | faster senescence | N |

**Table 2.** Phenological phases during which crop stress variables are used to compute the stress indices.

| Emergence | Maximum Acceleration of Leaf Growth | Maximum Leaf Area | Anthesis | Start of Grain Filling |
|---|---|---|---|---|
| | TempStressLUE | | | TempStressGrainFilling |
| | FrostStressPlantDensity | SenescenceWaterStress | | FrostStressReprod |
| | FrostStressFoliage | - | - | SenescenceWaterStress |
| - | | StomatalWaterStress | | |
| | | NitrogenLAIStress | | |

Water limitations were considered by their influence in reducing transpiration and radiation use efficiency (StomatalWaterStress) and for their induction of earlier plant senescence (SenescenceWaterStress).

Nitrogen limitations were considered taking into account three nitrogen stress variables concerning the optimal plant development and leaf growth (NitrogenLaiStress), the radiation use efficiency (NitrogenSenescenceStress), and the anticipated leaf senescence (NitrogenBiomassStress). However, as these variables had an almost identical temporal variation and were showed to be highly correlated on a principal component analysis (PCA), only one of them was kept for further analysis (Tables 1 and 2, Figure S3).

The Hi-sAFe stress variables were computed at daily time step and took values comprised between one (no stress) and zero (full stress), so that they could be used as reduction factors for state variables representing physiological processes. In order to obtain a more intuitive representation of the impact of stresses on plant performance, stress indices were computed as the complement to 1 of the mean annual values of the stress variables, and then multiplied by the number of days the same stress was experienced during the year (Equation (1)), resulting in increasing values for higher stress impacts.

$$\text{Stress index}_{\text{year}} = \left(1 - \text{mean}\left(\text{stress variable during selected period}_{\text{year}}\right)\right) \times \text{nb days of stress}_{\text{year}} \tag{1}$$

To better represent the trends in climate variables, annual crop yield, mean PAR intercepted and stress indices, the corresponding data series were presented after the application of moving averages (11 years and 11 pseudo-replicated series), while their variability were shown by a confidence interval applied over the same moving window.

However, as the windows contained some replicated years, the computed confidence intervals were not rigorous, but they were presented to give an indication of the variability introduced by the reshuffling process. In the resulting visualization, discontinuities arose at the limits between scenarios (in years 1990–1991 and 2030–2031): this was due to the fact that our moving averages did not take into account the values outside the time period.

## 3. Results

### 3.1. Example of Crop Yield Responses to Water Availability

We first illustrated the simulated tree–crop–environment interactions by analyzing the crop and tree developments during two contrasted consecutive years. These were chosen to be a "wet year" (total rainfall = 1229 mm) and the following a "dry year" (total rainfall = 557 mm) (Figure 2), with medium-sized trees that already had a significant impact on the crop (about 15-year-old trees, with 20 cm diameter at breast height, DBH).

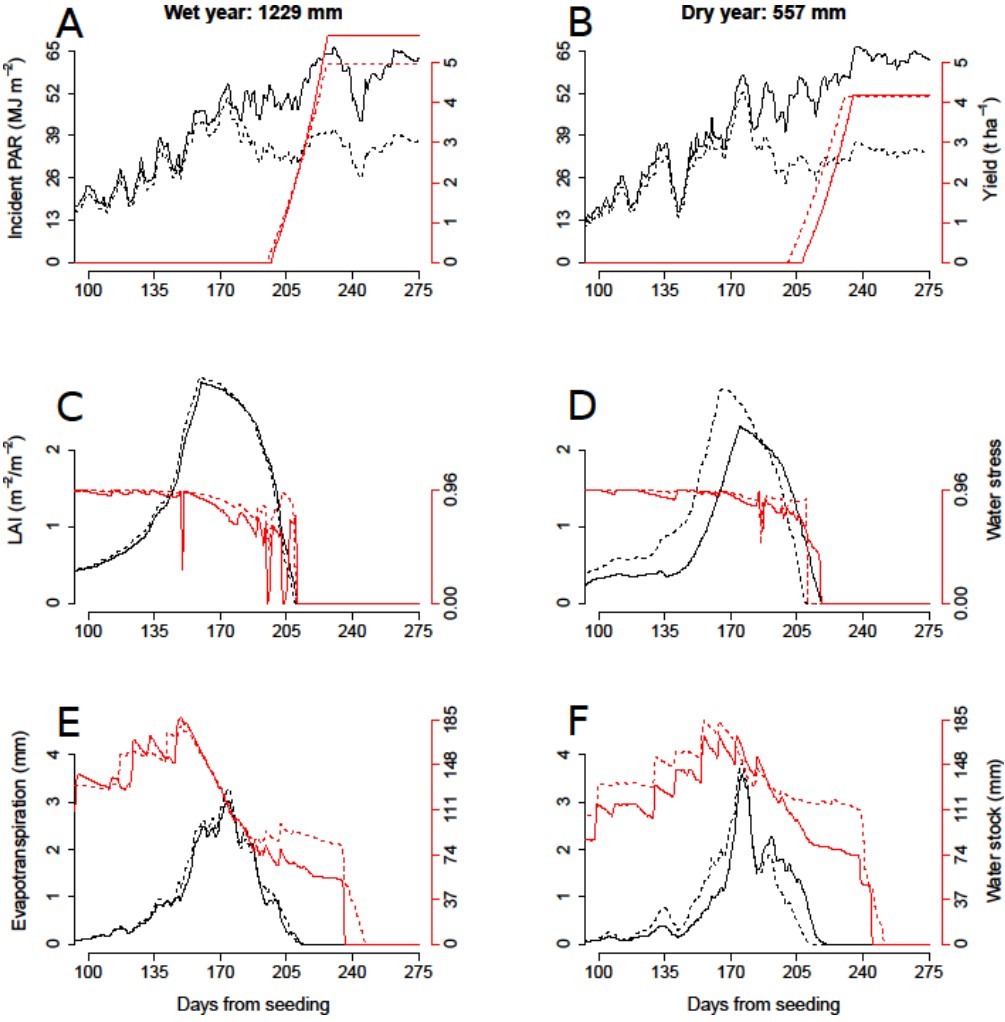

**Figure 2.** Physiological drivers of crop yield in a wet (**A,C,E**) and a dry year (**B,D,F**) across systems (solid lines represent monoculture, dashed lines represent alley cropping, black lines correspond to the **left** vertical axis, red lines to the **right** ones). (**Top**) Photosynthetically active radiation (PAR, black) incident on crops and crop yield (over the whole plot surface, red); (**middle**) crop leaf area index (LAI, black) and crop stomatal water stress index (red) ranging from 1: No stress, to 0: Full stress; (**bottom**) crop evapotranspiration (black) and water stock available for crops in the crop root zone (red). Values are given at plot scale. Left vertical axis is for black lines, right vertical axis is for red lines. For a clearer graphical representation, the PAR incident is shown after averaging with a moving window of 7 days.

At this age, leafless trees intercepted a small amount of light in the early season (until leaf development, at about DOY = 180), so that the crop reached full leaf expansion without important losses in intercepted PAR (Figure 2A,B). However, the tree cover already mitigated the microclimate from early season stresses (Figure 2C,D; difference between dashed and solid red lines), favoring higher crop leaf area index (LAI) development in AF in respect to A, in particular during the dry year (Figure 2D, black lines). Then, tree leaves started intercepting a significant amount of global radiation, reducing the incident PAR on the crop in AF in respect to A (black lines in Figure 2A,B, after DOY 180). However, because of different microclimate conditions, crop transpiration remained similar in both systems. In this respect, it is interesting to notice that the crop in the AF system had higher water use with respect to A in the early season of the dry year (Figure 2F). This was somehow compensated by the higher water use in A when moving into summer. The water stock in the soil accessible by crop roots was similar in both systems in the wet year until the maximum crop LAI was reached (and tree leaves started expanding), then decreased faster in A than in AF (Figure 2F). The crop yield in AF was 13% lower than in A in the wet year (Figure 2A). Conversely, the reduced crop water stress resulted in equal yields per hectare in the dry year (Figure 2B). The increase of crop yield per sowed surface in AF compensated the reduced cropped area in AF.

### 3.2. Crop Stresses across Agricultural Systems

The increase in minimum temperatures (Figure 1) resulted in a major reduction of frost impact on plant density, leaf area (Figure 3A,B—left), and too-cold sub-optimal temperatures for photosynthesis (Figure 3C—left). Frost stresses (especially on leaves) were predicted to be less intense than most other stresses. However, these stresses impact the crop in its early development, and their effects on final yield may be larger than indicated by their values. The decreasing trend in frost stresses presented an anomaly around year 15 in both *Past* and *Present* (Figure 3A,B—left), corresponding to the occurrence of particularly cold years during the second decade of the *Past* scenario, and the absence of warm years in the second decade of the *Present* scenario (Figure S2). The cold stress on the crop leaf area also had the effect of damaging the leaf area and indirectly reducing the intercepted PAR (Figure S5).

The rise in maximum air temperatures resulted in increased high-temperature stresses on photosynthesis (HighTempLue stress) and grain filling (TempGrainFilling) (Figure 3D,E—left). This was true when comparing *Past*, *Present*, and *Future* and within each scenario, except during the second half of *Future*. These stresses had relatively high intensity and increased twofold from *Past* to *Future*, with the same anomaly due to a few cold years around year 15 in *Past* and *Present* (Figure 3D,E). In addition, some very dry years (with either high wind or low rain) occurred around year 25 in *Past* and *Present* (Figures 1 and S2), and caused a local peak in these stresses (Figure 3D,E—left).

Precipitation was strongly associated with three crop stresses considered (effect of lack of nitrogen on LAI development, and effect of water stress on stomatal closure and on leaf senescence) (Figrues 4 and S3). The nitrogen stress pattern followed the maximum annual rainfall with a 30% increase from early *Past* to late *Future*, but with irregular fluctuations within and across scenarios. The first major increase in nitrogen stress occurred in the period from about year 15 to 25 of *Past* (Figure 4C—left), during which precipitations both decreased in total amount and concentrated in few extreme events (Figure 1). *Present* started with higher nitrogen shortages and maximum annual precipitation; both the index and the meteorological variable followed a mild reduction until year 15, then increased until the end of the scenario, corresponding to an increased nitrogen stress. The higher occurrence of nitrogen stress at the beginning of *Future* and its decrease throughout this scenario (moderate at its end) mirrored again the decrease in maximum annual rainfall.

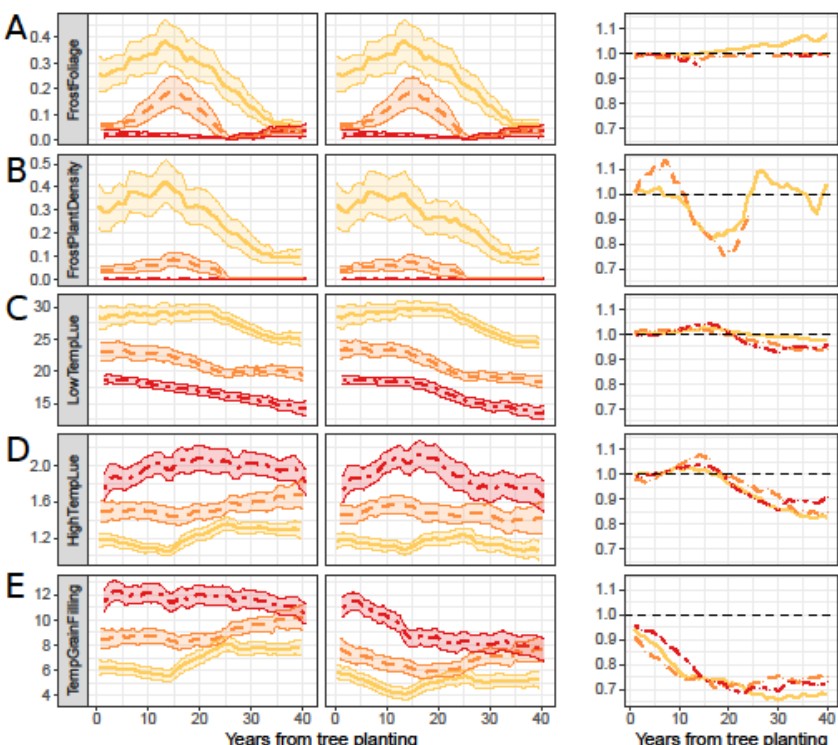

**Figure 3.** Crop thermal stress indices in agriculture (**left**), agroforestry (**middle**), and relative values (AF/A, **right**), in *Past* (yellow), *Present* (orange), and *Future* (red) climates. To present only the most interesting relative stress values, those resulting from absolute values lower than 5% the maximum stress value are not shown. Thermal stresses were: FrostFoliage for the effect of frost on foliage (**A**), FrostPlantDensity for the effect of frost on plant density (**B**), HighTempLUE, for the effect of high temperature on LUE (**C**), LowTempLUE for the effect of low temperature on LUE (**D**), and TempGrainFilling for the effect of high temperature on grain filling (**E**). Each scenario is represented by averaging 11 repetitions, while using a moving window of 11 years. Vertical lines represent the 95% confidence intervals.

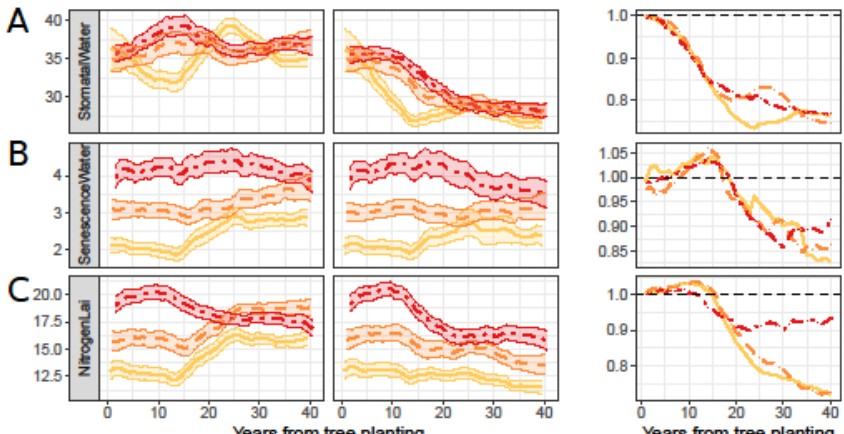

**Figure 4.** Crop nitrogen and water stress indices in agriculture (**left**), alley cropping (**middle**), and relative values (agroforestry systems (AF)/agriculture (A), **right**), in *Past* (yellow), *Present* (orange), and *Future* (red) climates. NitrogenLai for nitrogen stress inducing delayed plant development and slower leaf growth (**A**), SenescenceWater for stress inducing faster senescence (**B**), and StomatalWater for stress, inducing reduced radiation use efficiency and transpiration (**C**). Each scenario is represented by averaging 11 repetitions, while using a moving window of 11 years. Vertical lines represent the 95% confidence intervals.

The stomatal stress affecting radiation use efficiency and transpiration (StomatalWaterStress, Figure 4A—left) was one order of magnitude higher than the stress inducing leaf senescence (SenescenceWaterStress, Figure 4B—left), and had the highest values among all stresses.

StomatalWaterStress fluctuated over time, but without an overall increase across scenarios. High peaks (around year 20–30 in *Past*; years 10–20 in *Present* and *Future*) were associated to years with very low precipitation and high wind speed, and low peaks (year 13 in *Past*, year 25 in *Present* and *Future*) to colder temperatures and/or more uniformly distributed and higher rainfalls.

SenescenceWaterStress had a more stable and monotonous variation and an overall twofold increase over time, with exceptions in late *Past* and *Future,* mirroring stagnations in temperature and global radiation. Local variations in this index occurred concomitantly with the dry years occurring around year 25 in *Past*.

### 3.3. Crop Stresses in Agroforestry

Most plant stresses (HighTempLUE, LowTempLUE, TempGrainFilling, NitrogenLAI, SenescenceWater, and StomatalWater, Figrues 3C–E and 4A–C—middle) decreased with tree age in AF. The higher reduction was observed for the temperature stress on grain filling (down to minimum values of about 65% in respect to A). However, while some of them (temperature stress on grain filling, water stress on radiation use efficiency and transpiration) decreased in respect to A continuously during the whole tree growth, others (temperature on photosynthesis, nitrogen, and water on senescence) had minor relative increases during the first 15 years (up to about 10%). Frost stresses had similar values across agricultural systems.

The relative values of stresses were generally consistent among scenarios (Figures 3 and 4—right), with one major exception. The nitrogen stress in AF decreased to about 70–75% of the values in A by the end of *Present and Future*, but not in *Past* (Figure 4C—right). In this scenario, the relative impact of nitrogen stress decreased only until it reached a value of about 90% around year 20, when both the RainfallTot and RainfallMax started decreasing, thus likely lowering their pressure on nitrogen leaching in both A and AF systems.

### 3.4. Crop Radiative Regime

The PAR incident on the crop in A was almost constant over time, with a minor but steady increase over the three scenarios (Figure S6). Conversely, the PAR intercepted by the crop in A increased by about 10% over each of the *Past* and *Present* scenarios, while it decreased by about 4% over *Future* (Figure S5—left) as a consequence of leaf area variations.

The intercepted PAR in AF followed a monotonous decrease with tree growth, down to 70% of the values of A after 20 years, and ending up with slightly higher values in *Future* and *Present* than in *Past*, at about 40 to 45% of their initial values (Figure S5—middle). Intercepted PAR relative values of the three scenarios overlapped almost completely (Figure S5—right), suggesting that, while for each system the plant response in terms of LAI development differed across scenarios, this response was very consistent across systems.

### 3.5. Crop Growth

Crop yields had important interannual fluctuations (Figure 5; for tree growth see Figure S7), ranging from total crop failure to a productivity of 6.4 t ha$^{-1}$ in A (mean in *Past* = 4.07 + −0.89, *Present* = 4.57 + −0.86, *Future* = 4.93 + −0.68 t ha$^{-1}$) and from near total crop failure (0.1 t ha$^{-1}$) to 5.8 t ha$^{-1}$ in AF (mean in *Past* = 3.31 + −0.60, *Present* = 3.79 + −0.63, *Future* = 4.03 + −0.68 t ha$^{-1}$). The reduction in AF productivity in respect to A (overall 18%) was related to the reduced cultivated surface (−8% occupied by the tree rows) and to a decrease in yield in the shade of the trees.

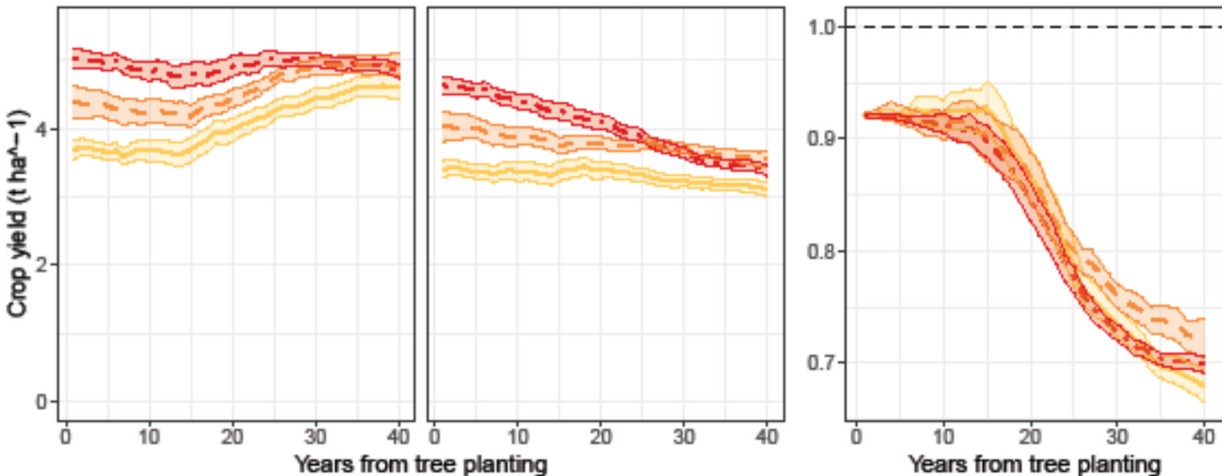

**Figure 5.** Simulated crop yield at plot scale in agriculture (**left**) and agroforestry (**middle**) systems and relative values (AF/A, **right**) in *Past* (yellow), *Present* (orange), and *Future* (red) climate. Each scenario is represented by averaging 11 repetitions, while using a moving window of 11 years. Vertical lines represent 95% confidence intervals.

The crop yield increased with CC over the 120 years in A, from the beginning of *Past* until the beginning of *Future*, from which it remained somewhat stable. Yields decreased in AF during the tree growth cycles, but more in *Future* than in *Past* and *Present*, and less during the first twenty years (total mean loss of 0.25 t ha$^{-1}$) than in the rest of the tree growth cycle (total mean loss of 0.43 t ha$^{-1}$). Yield variability was generally reduced in AF in respect to A in both absolute and relative (i.e., coefficient of variation) terms (Table 3). Large-sized trees in AF reduced the occurrence of particularly low- or high-yielding years (Figure S8).

**Table 3.** Yields by agricultural system, climatic scenarios, and typical parts of the 40-year rotation (start, middle, end). Cumulative yield is reported for years 5, 22, and 40.

| | | Yield (t ha$^{-1}$) | | | | | | | |
|---|---|---|---|---|---|---|---|---|---|
| | | **A** | | | | **AF** | | | |
| | | **Mean** | **SD** | **CV** | **Cumulative Yield** | **Mean** | **SD** | **CV** | **Cumulative Yield** |
| *Past* | Years 1–5 | 3.65 | 0.56 | 0.15 | 18 | 3.36 | 0.52 | 0.15 | 17 |
| | Years 18–22 | 4.16 | 0.83 | 0.20 | 83 | 3.49 | 0.57 | 0.16 | 75 |
| | Years 36–40 | 4.62 | 0.76 | 0.17 | 163 | 3.11 | 0.44 | 0.14 | 132 |
| *Present* | Years 1–5 | 4.38 | 1.08 | 0.25 | 22 | 4.03 | 0.99 | 0.25 | 20 |
| | Years 18–22 | 4.47 | 0.62 | 0.14 | 95 | 3.75 | 0.47 | 0.13 | 85 |
| | Years 36–40 | 4.95 | 0.70 | 0.14 | 183 | 3.55 | 0.46 | 0.13 | 151 |
| *Future* | Years 1–5 | 5.06 | 0.62 | 0.12 | 25 | 4.65 | 0.57 | 0.12 | 23 |
| | Years 18–22 | 5.06 | 0.64 | 0.13 | 108 | 4.15 | 0.48 | 0.12 | 96 |
| | Years 36–40 | 4.84 | 0.41 | 0.09 | 197 | 3.38 | 0.33 | 0.10 | 161 |

## 4. Discussion

### 4.1. Crop Stresses and Climate Change

In our simulations, extreme climatic events had a major impact on the decadal trends in crop stresses. The impact of a climatic variable on a crop stress was nonlinear, as it applied only when the variable crosses defined thresholds. This explained why a relatively small variation in a climatic driver may be responsible for severe physiological stresses and impacts on crop yield.

CC induced higher thermal [50–53], water [54], and nitrogen stresses in A (Figures 3 and 4) than in AF. Trees started mitigating most stresses after they had reached a sufficient size: high temperature stress on grain filling and water stress on radiation use ef-

ficiency were reduced for DBH lower than 10 cm (10-year-old plants); water stress inducing earlier senescence and nitrogen stress on leaf growth were reduced starting with DBH of about 20–30 cm (15–20-year-old plants); and high temperature stress on leaf photosynthesis were reduced starting with DBH of about 30 cm (20-year-old plants) (Figure S7).

The higher temperatures occurring in *Present* and *Future* scenarios, in respect to *Past* (Figure 1), were associated with higher heat stresses (Figure 3). AF combined a lower incident radiation on the crop (Figure S6), and reduced crop thermal stresses (Figure 3C–E). The heat stress reduction was particularly pronounced during the grain filling phase, even with young trees (<10 years), when the reduction in crop yield was still moderate or absent. For winter wheat, grain yield reduction during anthesis and maturity had already been reported for temperatures above 15 °C [50]. Before anthesis, temperatures above 31 °C were related to reduced grain number [50]. Similarly, in Chinese Spring wheat cultivar, relatively high temperatures (24 °C for a duration of five days) were associated with decreased floret fertility, dropping to zero for mean daily temperatures of 35 °C for five days [51]. In this respect, the AF system helped reduce the high-temperature stresses compared to A, and we may expect this effect to be even more substantial in field conditions warmer than our experimental site, such as in much of the Mediterranean agricultural areas or in continental climates.

High mean temperatures and extremely warm events between heading and maturity were also found responsible for early leaf senescence and reduction of the wheat growth cycle length [50,52,55]. Results from our simulations confirmed that the alley cropping system may favor later crop senescence and longer crop growth (Figures S9 and S10), as previously reported for wheat–Faidherbia alley cropping in semi-arid regions [30,55,56], and wheat–olive [30,55,56] and wheat–walnut [30,55,56] in Mediterranean regions. Despite the crop in AF having a longer life-span (anticipated emergence, delayed senescence), the start of the grain filling period did not change (Figure S11) [31], and the negative impact of high temperatures on crop development was reduced.

In our study, the total annual rainfall had important decadal variations with direct impact on crop water stresses, and indirect impact on nitrogen and other stresses. Rainfall occurred in fewer but more intense events, resulting in more water lost through runoff. The total yearly rainfall was stable or slightly decreasing. Both aspects, combined with higher temperatures, resulted in a significant increase in water stresses.

Water stresses were substantially reduced in AF, which should enable a higher efficiency in converting the available energy into crop biomass and yield [54]. The stress with highest intensity (StomatalWaterStress) was also the most reduced in absolute terms, even with young trees.

A key factor for avoiding tree–crop competition for soil water is complementarity in phenology and spatial arrangement [29,57]. In other alley cropping systems (apricot–millet, –peanut, and –sweet potatoes) complementarity allowed water consumption of intercrops to be lower than those of sole crops [57]. The wheat–walnut association presented a temporal lag between winter crops and trees in leaf formation and water use. In addition, due to a deeper root system, the water volume available for the AF system may be higher in respect to A, further reducing the possibility of water stress to occur.

Our study site was characterized by a deep silt–clay soil with high water capacity, and a river located nearby, resulting in water table four to eight meters deep. These conditions may have eased the access to water, preventing more pronounced water stress from occurring. Simulations, analysis, and model testing on larger sets of sites with different soil and environmental conditions would be useful to assess the suitability of this AF system in a broader context, and to predict its mitigation potential facing CC for dryer soil conditions. Periods with intense precipitation events were associated with increases in nitrogen stress in A (Figure 4C), as a consequence of the likely increase in nitrogen leaching. This process was suggested by [58] or [59] and was confirmed in our simulations. The AF system significantly reduced the impact of these extreme rainfall events (during *Past* and *Present*, down to 70% of the values in A), especially during the second part of the AF cycle,

when the trees were large (Figure 4C). This is a clear, though indirect, example of how the presence of trees in alley cropping may counterbalance the impact of extreme climatic events. A possible explanation for this phenomena is the role of tree roots as a "safety net", catching nitrogen leached below superficial soil layers [60]. Another explanation of the simulated process is that trees dry deep soil horizons in summer, resulting in a lower water content in deep layers in autumn, when mineralization rates are high in the humid top soil. N leaching out of the rooted zone is therefore reduced, giving more time to the trees and other plants (weeds) to capture more nitrogen in autumn [58], before it reaches the water table and is possibly lost from the plot. This nitrogen is recycled through tree leaf and root litters and remains available for the crop of the next year. Further analysis of the three dimensional model results, performed at finer temporal and spatial scales than in the present study, may help verifying this last hypotheses.

### 4.2. Crop Yield

The range of simulated crop yield in A in *Past* (3–6 t ha$^{-1}$) is consistent with yield values observed for Southern France [6], projected by other studies [61], and similar to field measurements (3.5–6.2 t/ha) performed in the same experimental site [43]. Whether wheat yields facing CC will increase in response to carbon dioxide fertilization [62] (effect not considered in our simulations) or decrease because of other factors such as droughts [61] is debated. The increase in crop yield simulated in A over time (from 3.65 t ha$^{-1}$ in *Present*, up to 5.06 t ha$^{-1}$ in *Future,* Figure 5) was explained by a decrease in the early season frost stresses and by less cold temperatures suboptimal for photosynthesis. These unfavorable conditions may be particularly related to the specific study site, which is located in a topographical depression, where winter temperatures are lower than in the nearby plain.

The Hi-sAFe model well predicted tree growth (Figure S7) [43], allowing us to analyze the microclimatic impact of walnut trees on crop yield. Crop yield in AF in *Past* was reduced by only 16% in respect to A, 20 years after tree plantation. This was the same reduction observed with only 12-year-old trees in the field [43], indicating that the model underestimated the impact of the trees on the crops. Previous studies on the same crop–tree association exist but, to the authors' knowledge, only with quite different plant densities. The simulated crop yields' reduction in AF was modest compared to results obtained in Western Spain [31] where a drop of 47% in crop yield was evidenced in a denser (+390%) but younger (9-year-old) plantation of the same species. In both systems, tree and crop roots tended to occupy different soil layers, but were still competing for nutrients and soil water, resulting in slower root growth and yield losses [43,63].

The decrease in PAR reaching the crop due to tree canopy interception (Figure S6) was an important driver of yield loss in AF systems [43,50]. This mechanism was less effective with hybrid walnut, as the shade of this late leafing tree was only significant after the crop maximum leaf area was reached [31,43]. This is a key advantage that explains the low reduction in crop biomass in the shade of walnut trees [43]. In addition, as a consequence of the asymptotic shape of the photosynthesis curve (e.g., for wheat see [63]), lower radiation intensity may be partly compensated by an increase in light use efficiency, as previously reported for winter wheat under artificial shading and in alley cropping [43,64,65].

### 4.3. Limitations of the Modeling Approach

As with any model, not all processes are fully represented in Hi-sAFe. Considering the water use, it is worth mentioning that the water demand of both trees and the crop is simulated as proportional to the potential evapotranspiration and the fraction of the light intercepted, without taking into account for leaf clumpiness, though this was considered as an acceptable simplification by [66]. In addition, the calculation of the potential evaporation for crops in the shade of the trees did not take into account interactions between wind intensity and the stand structure, as this computation was considered too time-consuming and difficult to parameterize [42]. Concerning tree growth, the tree response to nitrogen limitation was poorly parameterized and the effect of tree nitrogen stress on tree growth

was turned off in the current simulations. This was not considered as a limit for this study: we did not want to predict the accurate growth of the trees in this site, but to explore how large trees would impact the crop's resilience to climate change. Concerning indirect consequences of CC, biotic stresses (such as pests) are difficult to predict [36] and are not included in the model. Some impacts of climate events on crop yield were also not simulated, such as the higher crop lodging during extreme wind and rain events [67]. We may expect that the tree stand in AF could attenuate these effects due to wind speed reduction and increases in rain interception in respect to A.

## 5. Conclusions

Simulations with the Hi-sAFe model show that the complementarity for light use was high between hybrid walnut trees and winter wheat cultivated in Southern France, and resulted in a limited reduction of wheat yield until the trees reached a large size. CC surprisingly increased wheat yield in A in our site, as a consequence of lower frost or less cold suboptimal air temperatures in winter and early spring. In AF, walnut trees mitigated many important crop biophysical stresses, resulting in a less variable wheat yield than in A. CC is predicted to increase physiological stresses linked to high temperatures, but to decrease low-temperature-induced stresses. Large trees reduced high-temperature stresses, in particular during grain filling, as well as nitrogen stress on LAI (likely due to a reduction in nitrogen leaching) and water stress on stomatal closure. These effects only partially compensated for the reduction of intercepted PAR and led to a limited decrease of crop relative yield as trees grew, and a better yield stability in AF than in A. The complexity of interactions between climate variables, the timing of different physiological stresses, and their final impact on crop yield make it difficult to design appropriate field experiments, and emphasize the usefulness of simulation models that are able to predict the outcome of the competition/complementarity/facilitation balance in the tree–crop interactions for different soil and climate conditions. Based on our results, we can hypothesize that agroforestry would be more useful to mitigate the negative impacts of climate change on crops in situations where high temperature and water stresses would be damaging to crops. Nonetheless, particular attention has to be paid to the interaction with soil water availability, which could drastically limit tree growth and thus reduce the mitigation potential of trees on crop stresses.

**Supplementary Materials:** The following are available online at https://www.mdpi.com/article/10.3390/agriculture11040356/s1, Figure S1: Procedure for microclimatic adjustments, Figure S2. Pseudo-replicates of the climatic series, Figure S3. Principal component analysis of yearly climatic variables and stress indices in agriculture, Figure S4. Extreme temperature and precipitation during the year and across *Past*, *Present* and *Future* scenarios, Figure S5. Mean daily PAR intercepted by the crop since tree plantation in pure culture and alley cropping, and relative values (AF/A), across scenarios, Figure S6. Mean daily PAR incident on the crop since tree plantation in pure culture and alley cropping across systems and scenarios, Figure S7. Simulated tree DBH (cm) (above) and stem volume ($m^3$ $ha^{-1}$) (at plot scale, below) in alley cropping systems over time, Figure S8. Yield distribution at plot scale, across scenarios and agricultural systems, Figure S9. Variation in time of the day of plant emergence across scenarios and agricultural system, Figure S10. Variation in time of the day of start of plant senescence across scenarios and agricultural system, Figure S11. Variation in time of the day of start of grain filling across scenarios and agricultural system, Table S1: Linear models for the prediction of maximum and minimum relative humidity.

**Author Contributions:** Conceptualization, C.D., M.G. and F.R. Methodology and data analysis, F.R. Model development C.D., K.J.W. and I.L. Original draft preparation, F.R. Writing review K.J.W., C.D. and M.G. Funding acquisition C.D. All authors have read and agreed to the published version of the manuscript.

**Funding:** This research was funded by the European Commission through the AGFORWARD FP7 research project (Contract No. 613520).

**Institutional Review Board Statement:** Not applicable.

**Informed Consent Statement:** Not applicable.

**Acknowledgments:** We gratefully acknowledge the Fondation de France for supporting our agroforestry modeling efforts, and the Conseil Départemental de l'Hérault for allowing us to use the Restinclières Agroforestry experimental Farm Estate, where our models were calibrated and validated. We also acknowledge support of the European Commission through the AGFORWARD FP7 research project (Contract No. 613520). We also thank Guillaume Blanchet for fruitful discussions and suggestions.

**Conflicts of Interest:** The authors declare no conflict of interest.

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
