# Peer review of "Alley Cropping Mitigates the Impacts of Climate Change on a Wheat Crop in a Mediterranean Environment: A Biophysical Model-Based Assessment"

_agriculture, doi:10.3390/agriculture11040356_

Round 1

Reviewer 1 Report

General comment

The manuscript has been improved and most of the suggestions were adequately addressed. I must congratulate the authors for the work done. However, there are still some issues about which I would like to call their attention. The most relevant one is that I disagree with the consideration of crop yield as a second-interest variable. Crop yield is the ultimate and aggregative outcome (and the most relevant one for practitioners) of the stresses that you are describing and, in my opinion, should be more clearly emphasized. I agree on showing first the effects on the stress variables, as you have done so far, but I think that crop yield should be given more attention, as it is a result integrating all the other indicators. Likewise, I disagree with the removal of the reference to LER (Land Equivalent Ratio) in the manuscript, as it is a major (if not the most relevant one) indicator of agroforestry performance compared to agriculture. I would kindly request the authors to reconsider these decisions, as I assume that both yield and LER are variables provided (directly or indirectly) by the Hi-sAFe model and should be easy to remark / integrate. Otherwise maybe the title of the work should be changed, as “growth” may be interpreted as “yield”.

Moreover, I see that the Forestry model (F) is mentioned and described in the M&M section but there are no results or discussion provided on it, when it would be helpful for the reader. I would suggest describing it or otherwise removing any reference to this model.

The line numbers are not provided so I make my comments at section level, writing the whole sentence:

Introduction

- CC mitigation (e.g. carbon storage [24–26]  --> please use a closing parenthesis “)”

- “variables that are practically impossible to measure in the field" --> there are available techniques, although they are expensive. I would rather use “variables whose field assessment is considered unaffordable”

- Please change “walnut trees (Juglans intermedia (Carr.))” by “hybrid walnut trees (Juglans x intemedia (Carr.))”

Materials and methods:

- Please use always a space between each number and its unit (m) - this was done sometimes but not always – e.g. “Tree spacing was 13m”. Moreover there should be no space before a point (“ .”) or after a hyphen (“- “).

- The thinnings applied are not properly described. Do you consider thinnings in the F model (204 trees/ha)? What is the initial density and/or the thinning intensity in AF model? How do you achieve a regular layout (13x9 m) if the thinning you applied is selective? Shouldn’t it rather be a systematic / regular thinning?

- From 3.2 onwards, there are still errors in the reference sources (“Error! Reference source not found”). Please check and fix this problem.

Results

The results section should focus on depicting the results obtained, avoiding personal interpretations. For example, in 3.5 there are expressions that should be avoided such as “as usual in rainfed Mediterranean systems”; "surprisingly”, “as expected”. These considerations can be used in the discussion section, but not in the results section.

In Table 3 it would be useful having the overall yield after 40 years for each scenario, for a more comprehensive comparison of the aggregated effect of AF vs A.

Discussion

I think it would be more comprehensive to switch the order of 4.1 and 4.2, in order to keep the same structure as in the Results section (first stresses, then yield).

4.1 I don’t think it is meaningful to compare your results with those obtained in a system with 4 times more tree density. I would suggest using references of more comparable conditions.

4.2: “CC induced higher thermal [51,54–56], water [57] and nitrogen stresses in A (Figure 3, Figure 4), but trees in AF mitigated most of these stresses after they had reached a sufficient size” --> please try to be more specific, i.e. “during the second half of tree rotation / after 15-20 years”, etc. I assume that there is not a harsh threshold, but at least there should be a clearer way to give an estimation on when the trees start mitigating climate change effects in a consistent manner.

4.2. on the effect of trees on reduced impact from N leaching: I think that it should be stressed the effect of the tree root system as a “safe net” filtrating the lixiviates, which, in my opinion, has a more relevant effect on nitrogen cycle than (or at least it is compatible with) the reason provided- trees drying the soil, which, by the way, can have many unexplained implications in the system water balance and therefore on water stress. You can find several references on the “safe net” effect in the literature, with either alley cropping or riparian buffers.

Author Response

Dear Reviewer,

we thank you for your careful revision. Attached you may find our answers to your comments.

Kind regards.

Reviewer 2 Report

The authors present a simulation study on alley cropping and how it mitigates the impacts of climate change on crop growth in a Mediterranean environment. The text reads very well, and the topic is timely and absolutely relevant; a good catch for MDPI Agriculture. The analysis is a very interesting contribution to evaluating AF system, although I feel that the authors leave substantial model outputs unexplored. The shift in crop phenology, which in the introduction is mentioned as one important consequence of trees altering the temperature regime of the crop, is completely ignored in the analysis. The question, whether the later anthesis leads to a higher risk in heat stress is not answered, an neither is the question whether or not the prolonged grain filling period is beneficial for the yield. The authors should add this. Furthermore, the statements on nitrogen leaching are difficult to assess, as the reader is not well informed about how the model represents the N uptake mechanisms of crops and trees, the N mineralisation and the leaching processes. Therefore, the results are difficult to interpret. A flaw that the authors could easily repair.

I have found a number of other, minor issues, which I would ask the authors to address before resubmitting the manuscript.

P1. (Temperature increases have been related to crop …) . Asseng et al. 2015 is a relevant reference here. Nat. Clim. Change 5 (2), 143–147. https://doi.org/10.1038/nclimate2470.

P2. (Precipitation patterns [6] are expected to have major impacts on crop photosynthesis …) No, water availability in soil has an impact on the crop, causing drought stress. Drought stress leads to reduced crop photosynthesis. Precipitation patterns have an impact on water availability in soils. Please correct the causal relationship.

P2. (Precipitation patterns [6] are expected to have major impacts on crop photosynthesis … ff). The whole paragraph is odd. It is true that precipitation patters are expected to change and precipitation is projected to slightly increase in high latitudes. However, what is more important, is that at the same time temperature increase results in a four times higher evapotranspiration, which consumes soil moisture much faster during the season. Additionally, changes in high altitude streaming patterns lead to more stable pressure situations, causing a prolongation of no-rainfall periods, while in contrast aggravating rainfall extremes. These two are the major causes for increased drought risk in Europe. Please rewrite.

P2. (The key drivers of CC adaptation…). Please put the references to the statements where they fit.

P2. (… yields, especially in C4 plants.) Reference needed.

P5. (In particular we adjusted…) In particular, we adjusted…

P5. (Error! Reference source not found.) !!!

P5. Figure 1: Bad quality. Try to improve!

P5. (To explain yields in different agricultural systems and scenarios). Style: avoid starting a paragraph with a subclause.

P8. (In order to illustrate crop-environment …) Style: avoid starting a paragraph with a subclause.

P9. Figure 1: This should be Figure 2. Bad quality, try to improve!

P9. (Error! Reference source not found.) !!!

P10. (Precipitation was strongly associated…) The reader would need more information on how soil moisture affects the nitrogen availability of crop and tree in the model. I hope this comes up in the discussion: otherwise, it must be added to the M&M section.

P14 (Wester Spain) Western Spain

P15. (Trees dry deep soil horizons in summer, resulting in a slower water penetration in autumn, when mineralization rates are high. N leaching is therefore slower, allowing time to the trees and other plants (weeds) to capture more nitrogen before it reaches the water table and is possibly lost from the plot.) This needs a bit more explanation. In the zones where the trees exploit the water, soil organic matter turnover is very slow, contributing almost nothing to N release. N leaching is slower, yes, but in the absence of sufficient soil moisture neither crop nor tree can take up N. The N remains unused, until water infiltrates again. In principle, the effect on N leaching is the same in the end, but the process description is different. Please rewrite this paragraph and look out for a correct process description. The reader would also need substantially more information on how soil moisture affects N uptake of the plant in the model to understand this response.

Author Response

Dear Reviewer,

we thank you for your careful revision. Attached you may find our answers to your comments.

Kind regards.

This manuscript is a resubmission of an earlier submission. The following is a list of the peer review reports and author responses from that submission.

Round 1

Reviewer 1 Report

The paper deals with an interesting topic: agroforestry. However, the paper  shows some important deficiencies that makes difficult to follow it up. I will make suggestions for the authors:

1.- Introduction: It is vague with generalizations and some mixed sentences. (e.g. in the first paragraph, there is a mix of causes and consequences (temperature and CO2) that are equality treated. The importance of agroforestry and climate change is not provided  from a policy point of you (e.g. AGFORWARD results). Only CO2 is mentioned as GHG. Some sentences are not clear: what does mean "against increasing respiration costs"? There are also generalizations and contradictions like for example "Rising atmospheric carbon dioxide concentration is expected to positively impact plant water-use efficiency and productivity. However, ] showed that an increase in CO2 did not systematically result in an increase in yield when other limiting factors are at play." which are not dully explained. 
Some affirmations did not take into account the current use of alley cropping systems in the Mediterranean that has been successful for thousands of years (line 92-93). The sentence in line 95 to 98 is not clear, because experiments can be done with trees with different age/canopy cover to show results. The aim (line 108-109) is too broad. Moreover, the role of High safe is not provided, nor the previous results of this model or the comparison with other models.

Methodology: it is unnecesary complex.  It describes the model but not how can it be used, or where is the model availale. The description of the sresses are too broad and it is not clear, what is meant

Results: a description of clipick results are carried out, which is not the aim of the paper. This should have been included and better explained in the methodology. The rest of the results should compare the different types of land use and not each type of land use in a separate form.The results should not make an explanation to familiarize the reader with the interpretation of the results. The results should be self-explanatory for any reader. The tree component is not evaluated and the paper is mostly linked to the crop production. 

Reviewer 2 Report

Dear authors,

I find that this is a very interesting study, quite convenient in the current climate uncertainty context and with a solid modelling process. However, I find that some relevant improvements should be done to multiply its potential, as described below.

Abstract

Line 16: please use IPCC instead of IPPC

Line 17: it is not clear at this stage what the 3 scenarios (past, present, future) mean. The abstract should be self-explanatory

Line 22: please use “especially” and not “esp.”

Line 23: LER should be expressed with its full words and not as an acronym

Introduction

General comment: the presentation of climate context and impacts is too general. It should be focused on Mediterranean conditions, where the manuscript is addressed. I miss clear references to the projected climate impacts in the Mediterranean, and on their effects on Mediterranean agriculture (preferably focusing on the same species considered in the study). For example, the increase of temperatures can be beneficial to most crops in north Europe, but detrimental for most crops in south Europe.

Line 37: I would rather say “weather” than “climate”

Lines 42-43: please update this data and refer it to a publication / report

Line 67: I would change “For example, crop planting dates have been adapted to CC” by “For example, crop planting dates can be modified to adapt to CC”

Line 73. Please also mention the consideration of AF systems in EU regulatory framework, being specifically mentioned, for instance at the Green Deal

Lines 77-78: I would change “Increased available nitrogen and nitrogen cycling, thus reduced risk of nitrogen deficiency” by “Increased available nitrogen and improvement of nitrogen cycling, thus reduced risk of nitrogen deficiency and leaching”

Lines 74-84: the authors should support these arguments with references from studies conducted in Mediterranean conditions. One of the references is based on Canada, so their conclusions may not be applicable in the Mediterranean.

Lines 92-93: I am sure you can find various references supporting the fact that alley cropping is an appropriate CC adaptation strategy, e.g. Agroforestry as a tool to mitigate climate change” and “Agroforestry in Europe: a land management policy tool to combat climate change”

Line 98: “only a few of them currently exist” is not fully true. There are many experiments ongoing, with several publications produced. Nonetheless, it is true that modelling can help bringing forward many results and support decision-making

Line 111: I think “cycle” is better than “cycles”

Line 111: I think “Juglans x intermedia (Carr.)” is more widely used than “Juglans nigra x regia”)

Materials and methods:

Please use always a space between each number and its unit (m).

It must be mentioned whether an essential management intervention as thinnings are considered or not; if so, when are they applied and why did you choose these initial densities for the modelling.

Line 121. Please provide a more detailed indication of the geographical coordinates

Line 127-128 I assume that the grass on the tree line is not productive and simply maintained with a short development, is that right? It should be mentioned and described how this area is managed.

Line 130: I assume that the Hi-sAFe version used in this manuscript is an updated version of the first version released, isn’t it? The SAFE project ended in 2005.

Line 160: no need to mention Restinclières again

Line 165: please use “IPCC” instead of “IPPC”

Line 170. Please use “business-as-usual” instead of “business as usual”

Lines 167 and 174-176: It should be clarified why the historical data does not include the empiric weather data between 2005 and 2019/20 so that you can use 15 further years of real weather data. Moreover, I think it should have to be better explained why the authors are not using two climatic scenarios: past (1951-2020, with empirical data) and future (2020-70 with projected data). I see one problem of the current time periods, that is a strange evolution between 1990 and 1991 in Figure 1 (Results section), linked to the threshold between the scenarios.

Line 179: from here on, there is an error in the reference sources (“Error! Reference source not found”)

Line 180. Temperature adjustment should be more clearly explained: which years (both empiric and simulated) did you consider for your adjustment

Line 213-215: the sentence is not clear, please re-word it

Line 238: please say what LER stands for (Land equivalent ratio) and improve its description in line 239 (not meaningful)

Line 241. Please use “A or F” instead of “A and F”.

Results

In general I find that this section should be better structured to ease the understanding and coherence with the discussion section. I would suggest using a structure as follows: 3.1 Climate scenarios; 3.2 Crop yield, LER and radiative regime (integrating the current 3.6, 3.5 and a summarized version of current 3.2); 3.3 Crop stresses in agriculture and agroforestry (current 3.3 and 3.4). With this arrangement the authors could go from the most relevant information (yield) to the reasons behind it (stresses).

I also think that the usage of 2 scenarios (past and future) instead of 3, as suggested above, would simplify this section and would allow focusing on the most relevant impacts.

Finally, I miss references to the productive results of tree production, when comparing forestry with agroforestry system; if I understand right, in lines 145-146 it is stated that the forestry system was also modelled. These results would allow providing information on the LER, which is not mentioned in the Results section. There is only a minor explanation about this in lines 565-566 within the discussion, although I find that it would be a major benefit for the manuscript to include further information on tree development, even when it is not calculated in details. Maybe a simple indication of expected tree dimensions over time would be helpful.

I would suggest to re-think this structure for the chapter before going further into details on the specific content, for which I have only some preliminary comments:

Lines 247-252: this paragraph could be moved to M&M, where the model is described. I don’t think this is a result but rather a description of the model settings

3.1 I think this section should be summarized, I find it too long.

3.2. It is interesting to present the results of crop yield at approximately one third of the rotation (15 years) when having a wet and a dry year. However this should probably be done also at two thirds of the rotation (i.e. 30 years) so that the reader can have a wider understanding on the effect of trees. The resulting text should be presented more briefly.

This section 3.2 could be rather named as: “Example of crop yield responses to water availability” or “Example of main crop traits affected by water availability in various stages of tree development”, if the authors agree on providing also the situation when trees are aged 30.

3.3. If the information was organised in two scenarios (past and future), as suggested above, this section could be more easily explained. Moreover, the comparison of “past” scenario with the empirical data measured would make the model interpretation more consistent. Moreover I think that 3.3. and 3.4 should be merged. If I understand right, they deal with crop stresses in agriculture and agroforestry systems, and Figure 3 provides the results for both cases.

Line 335 “the impact of sub-optimal temperatures for photosyhtnesis” is not a clear sentence (sub-optimal because it is too low or too high?)

Figure 3. It should be clarified if the relative values refer to A:AF or AF:A. Moreover, for FrostPlantDensity it should be explained the strange trend of the future scenario between years 20 and 30, as well as the relative values graph (with values reaching 15x for the Present scenario and with the Past scenario unseeable (?). Can it be a miscalculation?

Discussion

I think that further emphasis should be done regarding the comparison between model outcomes and empirical data, particularly in chapter 4.1. I find that the main strength of this work is the possibility to validate / interpret the model outcomes referred to field data, even when it is not done in full details. Otherwise, the paper would simply explain how to use Hi-sAFe, which would be also useful, buy well below its potential.

I also miss references within the discussion to a key feature of hybrid walnut, which is a major advantage over other valuable broadleaved species in silvoarable systems: this hybrid has a late flushing that reduces notably the shading during spring. It should be explained: a) whether this is considered in HI-sAFe model or not; and b) the implications, according to the authors and previous experiences, of this late flushing on the study outcomes: less shading vs delayed temperature buffering effect.

Lines 470-474: there was no data provided on LER in the results section, so it should not be discussed here unless further information is provided.

I find this chapter 4.1 too short, considering that it is capital within this manuscript.

Line 497: this sentence should be written with more details

Lines 476-503: if I am not wrong (as there are many broken links it is not clear) there are no external references with which to compare the findings of this study. It would be helpful to know about what was found in previous studies, preferably in comparable conditions.

Line 518: there is only one reference, and it is related to semi-arid conditions. I would be interesting to some reference/s including conditions more similar to the study area

Lines 554-553: it should be further discussed what is the process by which the trees mitigate nitrogen stress and leaching: regulation of extreme precipitation events? Filtration of lixiviates with their root system? Nitrogen added to the soil as organic matter (leaves, twigs, fruits..)?

Conclusions:

I miss a clear statement comparing the crop yield in arable vs agroforestry system, before introducing the stresses affecting the yield.

Lines 580-583: once again, LER values are provided here but are absent in the results section.